# OpenReview forum: "$\mu$P for RL: Mitigating Feature Inconsistencies During Reinforcement Learning"
_ICLR.cc/2026/Conference — Submitted to ICLR 2026_

### Official Review · Reviewer_uKPN · 2025-10-30

**Soundness:** 3
**Presentation:** 3
**Contribution:** 3
**Rating:** 6
**Confidence:** 4

**Summary:**

This paper investigates how network parameterization affects learning consistency and computational efficiency in reinforcement learning (RL). It compares the Neural Tangent Kernel (NTK) parameterization with a proposed alternative, CompleteP, across a suite of continuous control tasks. The authors argue that RL’s non-stationary data distribution makes NTK scaling rules suboptimal, leading to instability when transferring hyperparameters across network widths and depths. CompleteP, in contrast, maintains more consistent feature learning and enables efficient scaling without costly hyperparameter sweeps. Empirical results on up to 16 continuous-control benchmarks show that CompleteP improves both reward efficiency and compute efficiency relative to NTK.

**Strengths:**

* *Timely and relevant topic*: Scaling laws and parameterization effects in RL are under-explored compared to supervised learning. The paper addresses an important gap with potential practical implications.
* *Comprehensive empirical analysis*: The authors evaluate multiple architectures, scaling factors, and optimization settings, and measure both reward and compute efficiency.
* *Clear empirical advantage*: CompleteP shows consistent improvements over NTK across tasks and widths, supporting its claim of better hyperparameter transferability.
* *Strong writing and organization*: The paper is clearly structured, with detailed appendices and well-designed figures summarizing trends in feature consistency.

**Weaknesses:**

* *Logical gap in motivation*: The paper attributes NTK’s failure in RL primarily to non-stationary data distributions, yet never establishes how non-stationarity causes NTK’s scaling rules to break down. Without a mechanistic or empirical link (e.g., gradient variance growth, kernel drift, or feature collapse under distribution shift), this reasoning remains speculative.
* *Mismatch between motivation and evidence*: The experiments focus on continuous-control benchmarks (mostly DeepMind Control Suite and MuJoCo), which feature relatively stationary dynamics and dense rewards. These settings do not strongly exhibit the non-stationarity that motivates CompleteP. Hence, the experimental results demonstrate better scaling efficiency, not necessarily stability under non-stationary data.

**Questions:**

* Could the authors clarify the precise mechanism by which RL’s non-stationary data affects NTK dynamics? Is the instability empirical (e.g., learning divergence) or theoretical (e.g., loss of kernel invariance)?
* Would CompleteP still provide benefits in explicitly non-stationary or multi-task settings (e.g. multi-agent environments), where environment distribution shifts are more severe?
* Are there diagnostic results (e.g., kernel alignment or feature CKA over time) that directly support the claim of improved “feature learning consistency”?
* How sensitive is CompleteP to optimizer choices or different scaling laws (e.g., under SGD versus Adam)?

---

> ### Author Response · Authors · 2025-11-21
> **Mismatch of claims, multi-agent environment**
>
> We thank the reviewer for the insightful comments. We detail our response below and refer to Figures, Tables and Appendices in the revised manuscript.
>
> **Logical gap:**  We appreciate the reviewer’s attention to this point and would like to clarify our intended claims. Our core assertion is not that non-stationary data causes only NTK parameterization to fail, but rather that RL’s non-stationarity makes it qualitatively different from supervised learning, where most scaling law results for NTK have been established. Our interest was to empirically investigate whether the benefits of parameterizations that encourage feature learning (like CompleteP) continue to hold in RL when using finite widths, given these differences. The observed instability and reduced feature evolution in NTK arise primarily from the kernel-like parameterization, not directly from non-stationary data. We do not claim that NTK’s scaling rules break down strictly due to non-stationarity—rather, our results indicate that limited feature learning (as induced by NTK) becomes a bottleneck as tasks become more challenging.
>
> We clarify in the manuscript "We clarify that the observed instability of NTK parameterization in RL arises from reduced feature learning capacity, rather than non-stationary data per se. While continuous-control tasks do exhibit evolving data distributions (see Fig. 13), our key finding is that enabling feature learning becomes increasingly critical as task complexity grows."
>
> **Mismatch:** We respectfully disagree with the characterization of our tasks as fully stationary. Fig. 13 (in the Appendix) illustrates that—even within standard continuous-control benchmarks such as CheetahRun and PandaPickCube—the data distribution (measured by its first four moments) shifts substantially, particularly during early training as the policy evolves. These dynamics reflect the inherent non-stationarity of RL. Furthermore, Fig. 34 and 35 (in the Appendix) demonstrates that CompleteP outperforms NTK in a sparse-reward setting, reinforcing its advantages where distributional shifts are more pronounced or where exploration is critical. While our primary results focus on reward and compute efficiency, the improved consistency we claim refers specifically to learning rate hyperparameter transfer and the speed of effective feature learning with CompleteP in comparison to NTK, not directly to the instability due to non-stationarity.
>
> **Multi-agent environment:** The main variable in our comparison is the degree of feature learning, governed by the $\Omega$ hyperparameter. Our results show that CompleteP maintains rich feature-learning capacity across scale, whereas NTK parameterization suppresses it as width increases. Our central hypothesis is that preserving rich feature learning improves learning stability and performance—especially important as RL environments grow in complexity or non-stationarity. At the same time, Fig. 10 and 11 (in the Appendix) and Graldi et al. (2025) note, excessive feature learning can lead to catastrophic forgetting and decrease in learning performance, suggesting that there is a nuanced trade-off. Extending this investigation to environments with more severe or longer-term distribution shift, such as multi-agent and non-stationary tasks, is an important direction for future work (as noted in our Discussion).
>
> **Diagnostic results:** Fig. 4 (left) show that CompleteP’s kernel alignment occurs at a consistent rate across different widths, unlike NTK, where alignment speed varies with scale. Fig. 4 (right) and Fig. 25 (Appendix) further demonstrate that kernel similarity between different seeds is higher and more stable (smaller variance) under CompleteP as width increases, suggesting improved consistency of feature evolution. While these results indicate a benefit, we acknowledge that the effect is moderate and do not claim a dramatic improvement, hence suggesting inconsistencies in RL rather than super-consistency given by CompleteP.
>
> **Optimizer sensitivity:** The compute efficiency and consistent improvement of learning curves is consistent when using CompleteP with SGD optimizer. However, agents trained using SGD optimizer demonstrated significantly lower performance (maximum reward and speed of learning) than when using Adam optimizer. This result is demonstrated in other RL studies, hence the use of Adam or RMSprop instead of SGD. Therefore, we did not investigate SGD further and focused on Adam optimizer for the manuscript instead.
>
> We hope our response addresses all of the reviewer's concerns, and warrants a score revision. Additionally, we look forward to additional discussions!

---

> > ### Comment · Reviewer_uKPN · 2025-11-27
> >
> > I appreciate the responses and efforts from the authors. My concerns have been solved. Therefore I raise my score from 6 to 8.

---

### Official Review · Reviewer_gqc6 · 2025-10-31

**Soundness:** 3
**Presentation:** 3
**Contribution:** 3
**Rating:** 6
**Confidence:** 2

**Summary:**

The paper investigates whether feature-learning parameterizations (CompleteP/µP with depth scaling) improve consistency and efficiency in RL compared to lazy (NTK) scaling. Using PPO on 16 continuous-control tasks/variants, the authors show (i) learning-rate transfer across width/depth with CompleteP; (ii) more consistent feature evolution and seed-to-seed policy alignment; and (iii) better compute and reward efficiency (Pareto frontiers) at large scales. They provide kernel/CKA analyses on a synthetic probe set and report results with multiple widths/depths and seeds.

**Strengths:**

- Learning-rate transfer across width/depth with CompleteP reduces sweep cost—useful for scaling PPO on hard tasks
- Feature consistency: width-independent feature evolution and higher seed-alignment under CompleteP (CKA/eigenspectrum evidence).
- Efficiency gains: clear Pareto analysis showing lower compute to reach the same reward and higher reward at fixed compute on challenging tasks (e.g., Humanoid Maze).
- Comprehensive scaling study: multiple widths (4→2048), depths, and tasks; includes sparse-reward variants and ablations like orthogonal init / layer norm notes.

**Weaknesses:**

- Compute metric simplification. “Compute = params × grad steps × env steps” ignores optimizer-dependent costs and simulator variance; wall-clock with hardware notes would improve fairness.
- Baselines & breadth. Little empirical contrast with Standard Param (SP) at scale or with mean-field/$\mu$P variants beyond CompleteP; vision/pixel-based tasks are left for future work. Minor typos (“ComplteP”), a few caption clarifications needed.

**Questions:**

- How sensitive are results to optimizer (Adam vs. SGD) given the parameterization tables? Any changes to the scaling rules alter conclusions?
- Can you include a small sweep to confirm NTK’s best LR per depth doesn’t close the gap, and a SP baseline at select widths?

---

> ### Author Response · Authors · 2025-11-21
> **New figures for walltime and small sweep for NTK and SP comparison against CompleteP**
>
> We thank the reviewer for the insightful comments. We detail our response below and refer to Figures, Tables and Appendices in the revised manuscript.
>
> **Compute metric:** We apologize for this simplification. We initially felt that optimizer-dependent cost and simulator variance would be similar across scales and parameterizations. Nevertheless, we agree that there maybe some deviations. We have now included Figure 45 in the Appendix that includes the wall clock versus reward for the environments Swimmer, G1rough and HumanoidMaze. In these environments, CompleteP demonstrates faster learning performance over the NTK parameterization over different widths for the same walltime. These environments were trained with a single GPU type (A100, H100) as opposed to a heterogenous mix for other environments.
>
> We have added a new **Figure 45** and revised the manuscript "Refer to Fig. 45 for learning performance against wallclock runtime."
>
> **Baselines and breadth:** With regards to baselines, Appendix Fig. 9d shows Standard Parameterization demonstrating worse learning rate hyperparameter transfer than NTK and CompleteP parameterizations. Based on your request, we performed a small sweep to confirm NTK's best LR does not close the gap to CompleteP (**Appendix U, Fig. 46**). Additionally, we see that only after tuning the LR for SP allows agents to achieve the same maximal performance as CompleteP (**Table 11 and Fig. 46**). Hence, incurring the hyperparameter compute overhead, which CompleteP negates. We clarify that the key bottleneck is the need to perform hyperparameter sweeps which incur significant compute resources, whereas CompleteP does not, supporting out claim about additional compute savings due to hyperparmeter transfer.
>
> Additionally, Yamamoto et al. 2025 gave a theoretical account on how a proposed mean-field RL variant with deep representation learning significantly improves value learning performance. At the current moment, we consider CompleteP to be the most complete parameterization for maximal parameter update across width and depth scaling.
>
> We clarify in the manuscript as "Refer to Appendix E for learning rate transfer experiments for other environments and Standard Parameterization."
>
> With regards to breadth, we kept our simulations to only continuous control tasks as a key application of end-to-end network training using RL is sim2real for robotics where we train agents end-to-end in simulations before transferring the agent to real world hardware. Vision based-tasks demonstrate improved benefits when using a pre-trained vision encoder and attach a policy network on top [1] or using a contrastive loss function for the vision encoder [2]. Hence, we focused our analysis on continuous control tasks. Since we observe that the benefits of CompleteP in Supervised Learning transfers to an extent to RL, we predict that the same benefits of training vision encoders using CompleteP in SL could also transfer to the RL setup.
>
>
> **Sensitivity to optimizers:** It is a common practice to use Adam optimizer instead of SGD for RL as the momentum improves the speed of policy convergence. Likewise, in our initial analyses, using SGD optimizer resulted in poor task learning for both CompleteP and NTK parameterizations, but with CompleteP demonstrating significantly better performance than NTK. Hence, we expect the same conclusions for SGD as what we observe using Adam optimizer, as the main difference would be the presence and absence of feature learning.
>
> [1] Lin et al. 2023 https://arxiv.org/abs/2309.04504
>
> [2] Srinivas et al. 2020 https://arxiv.org/abs/2004.04136
>
> We hope our response addresses all of the reviewer's concerns, and warrants a score revision. Additionally, we look forward to additional discussions!

---

### Official Review · Reviewer_VoM4 · 2025-11-01

**Soundness:** 3
**Presentation:** 3
**Contribution:** 3
**Rating:** 6
**Confidence:** 3

**Summary:**

This paper investigates the application of the CompleteP parameterization (a variant of μP) to reinforcement learning agents, demonstrating that it mitigates inconsistencies in hyperparameter transfer, feature learning, and policy evolution across model widths and depths. The authors empirically show that CompleteP enables stable learning rate transfer, consistent feature representations, and improved compute/reward efficiency on 16 continuous control tasks compared to the NTK parameterization, arguing for its adoption to enhance RL scaling.

**Strengths:**

- Investigates the use of μP/CompleteP in RL, tackling non-stationary data with empirical evidence of stable hyperparameter transfer and feature consistency.
- The claims are well-supported by extensive experiments on 16 continuous control tasks.

**Weaknesses:**

- Lack of Theoretical Justification for RL

The paper provides no new theoretical backing for why $\mu P$'s benefits, designed for stationary data, transfer so robustly to the non-stationary RL setting. It relies on borrowing the SL theory and providing empirical validation.

- Ambiguous definition of CompleteP

The method is introduced conceptually (“μP variant adapted for RL”) but without a formal scaling equation or algorithmic pseudocode.
It is unclear how its initialization and learning-rate rules differ from μP in exact terms.

- Questionable Necessity for Large-Scale RL

The paper assumes that scaling benefits from LLMs apply to RL, but RL often works with small networks—unclear if large models are frequently needed beyond niche cases like RLHF in LLMs.

- Minor Points

The paper contains presentation flaws, such as a missing reference "Appendix ??" on L172, which should be corrected.
OpenReview title (“μP for RL”) and PDF title (“CompleteP for RL”) mismatch.

**Questions:**

- Why focus on large-scale RL when most agents use small networks? How does this align with RL's sample inefficiency versus LLM scaling?
- Could you extend mean-field theory to prove CompleteP's advantages in RL's dynamic distributions?

---

> ### Author Response · Authors · 2025-11-21
> **Theoretical underpinning, and necessity for large-scale RL**
>
> We thank the reviewer for the insightful comments. We detail our response below and refer to Figures, Tables and Appendices in the revised manuscript.
>
> **Theoretical Justification for RL:** The theoretical reasons we suspect muP to be good for supervised learning (SL) (i.e. consistent feature learning across scales and a well defined stable feature learning limit) can also be certified in the RL setting. The open problem is really emprical, i.e. is muP possibly useful at reasonable widths in practical RL scenarios. Hence, we wanted to first determine the extent to which SL benefits may transfer to RL, due to the problems of stochastic action sampling, non-stationary data distribution and loss of plasticity. Despite these fundamental differences, we do see consistency, but only to some extent. We discussed that this consistency is only observed when we use sufficiently wide networks i.e. agents with extremely small widths N<16 do not demonstrate the consistency observed in SL partly due to the difference in reward dependent loss. Furthermore, theory to describe learning dynamics in deep RL is only begining to emerge (Bordelon et al. 2023 - mean-field theory for TD learning dynamics but in one layer network; Yamamoto et al. 2025 - mean-field theory analogue with modifications to RL to demonstrate the benefits of representation learning in deep RL). Hence, our future work is to develop a theoretically motivated explanation for why or why not we should observe improved consistency and learning performance using the insights and data collected from our empirical simulations. To be completely honest the success of muP in supervised learning isnt that well understood theoretically yet.
>
> We expanded the related works section to cover related theoretical works for RL: "theoretical work suggests that mean-field scaling could lead to richer feature learning and improved optimization, though these results have only been established for value learning, not policy learning \citep{yamamoto2024mean}. Hence, empirical validation of rich parameterizations (e.g. $\mu P$) within challenging RL domains remains limited."
>
> **Definition of CompelteP:** We apologize for the lack of clarity. The main difference between CompleteP (alpha=1) and muP (alpha=0) is the additional depth scaling constant (1/L^\alpha) introduced to scale the output of the residual network. The initialization and learning rates are scaled the same way as muP.
>
> We clarified in the manuscript "Note that the precursor $\mu$P parameterization uses $\alpha=0$, while scaling the other hyperparameters similarly to CompleteP."
>
> **Necessity for large-scale RL:** Hilton et al. 2023 (https://arxiv.org/abs/2301.13442), Rybkin et al. 2024 (https://arxiv.org/abs/2502.04327), Fu et al. 2025 (https://arxiv.org/abs/2508.14881v2) and our work demonstrated that scaling the model size of RL agents monotonically improves learning performance on hard tasks. Large-scale RL with good initialization could be a useful strategy to solve robotics i.e. sim2real, which is to first train artificial agents in physics constrained simulations e.g. continuous control tasks and subsequently transfer the pre-trained agent to real-world, hardware constrained setup. However, efficiently training large-scale RL is difficult, further motivating our work. We demonstrate that larger network agents achieve higher maximum reward than smaller network agents. Additionally, we demonstrate that while small networks maybe sufficient to solve (achieve maximum rewards) simple tasks e.g. Cartpole, they need to sample the environment more (larger number of environmental steps) compared to larger network agents. Hence, taking an LLM scaling approach for RL might be the way to solve autonomous agents for robotics. We have not seen many large-scale RL proposals yet as scaling RL is hard, and there are only a handful of hard simulations that can only be solved by scaling RL agents, which we explored in this manuscript e.g. G1, HumanoidMaze etc.
>
> We clarify in the manuscript "Recent research shows that increasing model width \citep{hilton2023scaling}, depth \citep{wang20251000}, or the updates-to-data ratio \citep{fu2025compute, rybkin2025value} consistently improves RL performance, resulting in higher rewards. Such advances are particularly vital for robotics, where closing the sim2real gap requires agents with strong generalization and robustness. However, the influence of model parameterization on feature learning remains poorly understood."
>
> **Minor corrections:** Thank you for pointing these out! We have corrected them and other presentations flaws.
>
> We hope our response addresses all of the reviewer's concerns, and warrants a score revision. Additionally, we look forward to additional discussions!

---

### Official Review · Reviewer_fdLM · 2025-11-01

**Soundness:** 2
**Presentation:** 1
**Contribution:** 2
**Rating:** 4
**Confidence:** 3

**Summary:**

This paper systematically investigates how Neural Tangent Kernel (NTK) and CompleteP parameterizations influence hyperparameter transfer, feature consistency, and policy consistency within the context of online Reinforcement Learning (RL). The authors demonstrate that adopting the CompleteP parameterization effectively mitigates scaling inconsistencies, resulting in consequential improvements in compute and reward efficiency compared to the lazy NTK regime across continuous control tasks.

**Strengths:**

- The authors conduct extensive experiments, consuming significant computational resources to compare the models' learning dynamics up to the point of achieving final rewards in various environments.
- Authors provide empirical evidence on the scaling effects of NTK and CompleteP in the non-stationary RL setting , where these parameterizations have been non-trivially studied before.

**Weaknesses:**

- The reliability of the consistency findings (e.g., parameter transfer, feature/policy consistency) is constrained as the detailed analysis and visualizations are often restricted to only one or two specific environments (e.g., HalfCheetah, HumanoidMaze, PandaPickCube). A more comprehensive, universally quantified summary across all tasks would strengthen the generalizability of the conclusion.
- The work primarily compares the two parameterizations (NTK vs. CompleteP) which are specific scaling theories. It lacks comparison against established baselines from the general RL community, such as methods designed explicitly to mitigate seed variance or enhance plasticity (e.g., using orthogonal initialization or layer normalization as a baseline in core experiments). This makes it difficult to ascertain the absolute benefit or difference in learning dynamics relative to standard industrial practices.
- The related work section is sparse regarding modern techniques for resolving RL inconsistencies. The authors should enrich this section.

**Questions:**

In terms of sample efficiency and compute efficiency, does training using the NTK and CompleteP parameterizations demonstrate superiority compared to other methodologies (i.e., standard RL practices like [1])?

[1] Lee et al, "SimBa: Simplicity Bias for Scaling Up Parameters in Deep Reinforcement Learning"

---

> ### Author Response · Authors · 2025-11-21
> **Clarification on reliability and comparisons to established baselines**
>
> We thank the reviewer for the insightful comments. We detail our response below and refer to Figures, Tables and Appendices in the revised manuscript.
>
> **Reliability of consistency:** We performed 1) learning rate transfer analysis for HalfCheetah and HumanoidMaze (shown in Fig. 3) as well as for CartpoleSwingup, AcrobotSwingup, PandaPickCube (shown in Appendix Fig. 9), and 2) feature/policy consistency analyses on HalfCheetah (Figs. 4 & 5) as well as CartpoleSwingup, AcrobotSwingup, PandaPickCube and HumanoidMaze in the Appendix (Figs. 15 to 24, and 35), including seed specific feature and logit analyses for AcrobotSwingup and PandaPickCube. This makes it a total of five different environments.
>
> We included a clarifying sentence in the manuscript "Refer to Appendix E for learning rate transfer experiments for other environments." and "Refer to Appendix P for feature consistency in other environments."
>
> **Comparison against established baselines:** We compared 1) learning rate transfer using Standard Parameterization (Fig. 9d learning rate transfer for HalfCheetah) and 2) learning performance using Orthogonal Weight Initialization (Figs. 29 to 32) and Layer normalization (Fig. 33) which are summarized in Fig. 8, Tables 9 & 10.
>
> We clarify in the manuscript "Refer to Fig. 9d and Appendix U for learning rate transfer using Standard Parameterization."
>
> **Standard RL Practices:** We demonstrate that CompleteP is complementary with layer normalization (Our models already use observation normalization and resnets), as suggested by [1], to demonstrate improved learning performance (Figs. 8, 32 and Tables 8 & 9).
>
> We included a discussion on the complementary nature of schemes: "Initializing network parameters using orthogonal initialization instead of Gaussian distribution improved NTK agent's learning performance for some environments (Appendix N.2}). Additionally, layer normalization has been shown to improve reward learning performance \citep{lee2025hyperspherical}. We noticed a mixture of improvements in NTK agents in terms of maximal reward that can be attained at lower compute budgets (Appendix N.3). Despite these techniques, CompleteP agents with orthogonal or layer normalization techniques and consistently demonstrated improved reward and compute efficiencies, suggesting that CompleteP parameterization can work synergistically with other methods to improve reinforcement learning performance. "
>
> **Sparse related works:** We reduced the related works section due to space constraints. We have expanded the related wroks section on techniques to reduce inconsistencies in RL such as 1) Bigger, Regularized, Optimistic (BRO), 2) layer normalization, 3) weight regularization etc.
>
> We included the following paragraph in Related Works: "\textbf{Techniques to mitigate inconsistencies.} A number of modern strategies have emerged to address inconsistency and instability in RL beyond simply increasing scale. Dormant neuron reinitialization \citep{lee2024simba, dohare2024loss} help restore plasticity by reactivating underutilized network units, combating the staleness that can arise during prolonged training. Layer normalization and orthogonal normalization techniques \citep{lee2025hyperspherical} have proven effective for stabilizing optimization and reducing variance across different seed runs. Moreover, explicit regularization e.g. weight decay, remains important for controlling capacity and improving generalization. The “BRO” (Bigger, Regularized, Optimistic) approach \citep{nauman2024bigger} combines larger models, weight regularization, and mechanisms to balance exploration and exploitation to further reduce performance variability and support consistent learning at scale."
>
> We hope our response addresses all of the reviewer's concerns, and warrants a score revision. Additionally, we look forward to additional discussions!

---

### Author Response · Authors · 2025-11-21
**Summary of response to reviewers**

We sincerely thank all the reviewers for their insightful feedback on our work.

The reviews highlighted several important concerns, including:
- The need for broader baseline comparisons and fairer analyses (i.e. wall clock) against established parameterizations (e.g. Standard Parameterization)
- Clarification on the role of feature learning in reinforcement learning

In response, we have updated the manuscript and performed the following additional experiments as requested:
1. We directly compare CompleteP, NTK, and Standard Parameterization (SP), where the best learning rate for each width is selected to maximize performance for NTK and SP while CompleteP uses the same learning rate (see Appendix U, Figure 47).
2. We replot learning performance using wall-clock time, rather than compute or environment steps, to provide a fairer assessment of algorithm efficiency (see Figure 46)
3. We expanded the Related Works section and rewritten parts of the manuscript to improve the clarity on CompleteP and references to related figures in the Appendix.

We hope these revisions and new results address all major concerns, and we look forward to further discussions during the rebuttal and decision process.

---

### Meta-Review · Area_Chair_m9Fc · 2026-01-07

**Summary:**

This paper consists of an empirical evaluation of the CompleteP parameterization, a variant of maximal update parameterization ($\mu$P), in the context of online Reinforcement Learning, the goal being to extend scaling results from supervised learning to the RL case. The authors compare CompleteP against the Neural Tangent Kernel across sixteen continuous control tasks. Results show that CompleteP can yield stable hyperparameter transfer, especially for learning rates, across varying network width and depth. This transferability significantly reduces the computational overhead typically required for hyperparameter sweeps. Furthermore, the study finds that CompleteP enhances compute and reward efficiency while promoting more consistent feature evolution and improved policy alignment across different random seeds. While NTK scaling may struggle with the non-stationary data distributions inherent to RL, CompleteP maintains "rich" feature learning, which the authors argue is critical for performance as task complexity and model scale increase.

The main issues contributing to the recommendation to Reject are:

+ **Limited empirical breadth**: Reviewers noted that detailed analysis and visualizations are limited to a few specific environments (e.g. HalfCheetah and HumanoidMaze) rather than a quantified summary across all tasks (fdLM). Additionally, the study did not include vision or pixel-based tasks (gqc6).
+ **Limited baseline comparisons**: The original submission lacked comparisons with standard RL baselines such as Standard Parameterization or modern techniques like orthogonal initialization and layer normalization designed to reduce seed variance and enhance plasticity (fdLM and gqc6).
+ **Lack of theoretical justification**: The paper provides no theoretical backing for why benefits designed for stationary supervised learning transfer to the non-stationary RL setting and relies entirely on purely empirical validation (VoM4).
+ **Clarity of presentation**: Reviewers identified a lack of formal scaling equations and algorithmic pseudocode for CompleteP, making exact initialization and learning-rate rules unclear (VoM4). Multiple reviewers identified a range of minor presentation flaws, such as missing references, typos, and a title mismatch between the PDF and the submission portal (VoM4, gqc6).
+ **Logical gaps in motivations**: A reviewer commented that the authors rely on a speculative link between non-stationary data and the failure of the Neural Tangent Kernel (NTK) scaling rules and that the paper never establishes a clear explanation (e.g. kernel drift or feature collapse) for why non-stationarity breaks (uKPN).
+ **Questionable necessity and benchmark mismatch**: It was noted that, although scaling is vital for Large Language Models (LLMs), many RL agents utilize small networks, and thus the need for RL parameterization is unclear outside of niche cases (VoM4). Additionally, Reviewer uKPN commented on the limited nonstationarity in the dynamics of the selected benchmarks from which derive the motivations for CompleteP in the RL case.

The critical point leading to the final recommendation is that **this paper represents a purely empirical analysis and provides no new technical or theoretical insight into the underlying problem of neural scaling laws for RL**. As such, the novelty of the contribution is limited (all of the evaluated approaches are adopted as-is from the existing literature) and, **due to the issues raised by reviewers, the potential impact of the empirical evaluation is similarly limited**. The recommendation is thus to Reject.

**Reviewer Concerns:**

The author rebuttal addressed issues related to comparison with established baselines with additional experiments using Standard Parameterization, Layer Normalization, and Orthogonal Initialization. Issues related to the breadth of empirical evaluations were addressed with new analyses on five different environments. The authors additionally provided new evalations of wall-clock timings to address questions about efficiency. These revisions partially address reviewer concerns, however the additional experiments constitute a significant revision of the original submission (the current revision contains 40 pages of supplementary material).

Issues related to methodological clarity and theoretical motivations were not adequately addressed in rebuttal.

**Reviewer Scores:**

+ **R1 (fdLM)**: Concerned with limitations in the empirical analysis (many limited to just two environments), omission of standard RL initialization benchmarks, and limitations of the related work section. Unclear if the reviewer opinion would have been influenced by the author rebuttal, but there does not seem to be enthusiastic support from the paper in the original review.
+ **R2 (VoM4)**: Concerned about a lack of theoretical justifications, clarity in technical descriptions, and questionable need for parameter scaling. I find it unlikely the reviewer would have been convinced to raise their score on the basis of the reviewer rebuttal, especially those related to theoretical justification and technical clarity.
+ **R3 (gqc6)**: Concerned about breadth of baseline comparisons and metrics used. This reviewer might have been convinced by the additional wall-clock timing evaluations, but unclear if the expanded baseline analyses would have been convincing.
+ **R4 (gqc6)**: Initially raised concerns related to logical gap between motivations and actual non stationarity in benchmarks, and mismatch between motivation and evidence. The reviewer stated they were convinced to increase score during the discussion period after a, frankly, not very convincing author rebuttal.

---

### Decision · Program_Chairs · 2026-01-26

Reject